# Bilateral Renal Large B Cell Lymphoma in a Dog: A Case Report and Review of the Literature

**DOI:** 10.3390/vetsci8110258

**Published:** 2021-11-01

**Authors:** Emmanouela P. Apostolopoulou, Ioannis Vlemmas, Dimitra Pardali, Katerina K. Adamama-Moraitou, Theofilos Poutahidis, Paraskevi L. Papadopoulou, Georgia D. Brellou

**Affiliations:** 1Department of Pathology, Faculty of Health Sciences, School of Veterinary Medicine, Aristotle University of Thessaloniki, 54627 Thessaloniki, Greece; emmaapos@vet.auth.gr (E.P.A.); ivlemmas@vet.auth.gr (I.V.); teoput@vet.auth.gr (T.P.); 2Diagnostic Laboratory, Faculty of Health Sciences, School of Veterinary Medicine, Aristotle University of Thesssaloniki, 54627 Thessaloniki, Greece; dpardali@vet.auth.gr; 3Companion Animal Clinic (Medicine Unit), Faculty of Health Sciences, School of Veterinary Medicine, Aristotle University of Thesssaloniki, 54627 Thessaloniki, Greece; kadamama@vet.auth.gr; 4Department of Diagnostic Imaging, Faculty of Health Sciences, School of Veterinary Medicine, Aristotle University of Thesssaloniki, 54627 Thessaloniki, Greece; vivipap@vet.auth.gr

**Keywords:** lymphoma, kidneys, canine, B-cell, neoplasia, immunohistochemistry

## Abstract

Canine lymphoma is a commonly reported neoplasia and, in most dogs, arises from lymph nodes before spreading to other organs. Renal lymphoma rarely occurs, and kidneys usually are a secondary site of origin. Primary renal lymphoma is infrequently described in the veterinary literature. In this study, we present a rare case of primary renal lymphoma in a dog and a review of similar cases. A 3-year-old male dog was admitted due to anorexia, weakness and vomiting. Clinical examination revealed bilaterally enlarged kidneys. Imaging demonstrated the presence of multiple renal masses. Cytology of abdominal fluid and kidneys led to the diagnosis of large cell lymphoma. Histopathology and immunohistochemistry on tissue samples taken from the kidneys confirmed the cytological diagnosis of lymphoma and categorized it as primary bilateral renal large B-cell lymphoma (LBCL).

## 1. Introduction

Lymphoma accounts for approximately 80% of all canine hematopoietic tumors [1,2]. The most common type is diffuse large B-cell lymphoma with lymph nodes being the primary site of origin. Primary extranodal lymphoma is less frequently described in dogs and is mainly of T-cell origin with poor prognosis [3,4,5,6,7,8,9,10]. Primary renal tumors are uncommon in dogs, comprising 0.2–1.7% of all canine neoplasms [1,6,8]. The majority of them are of epithelial origin, whereas very few cases of sarcoma or nephroblastoma have been reported [11]. Canine renal lymphoma is typically secondary and is usually identified as part of multicentric lymphoma [12]. In the veterinary literature, 16 cases of presumed primary renal lymphoma have been reported in dogs 2–9 years of age (Table 1) [1,12,13,14,15,16,17,18,19,20,21,22,23,24,25]. However, gross and histopathological examination was performed in only nine of them [1,13,16,18,19,20,22,23,25], while for the rest of the cases the information presented was insufficient to support the diagnosis. Sometimes lymphoma is reported or implied to be confined to one body system, but in many cases a thorough clinical examination or necropsy has not been performed to determine accurately how confined or not the neoplasm was [2]. To our knowledge only six of the 16 reported cases were investigated immunohistochemically and the neoplastic cells were identified as being of T-cell origin [12,16,17,19,20].

Regarding epidemiology, most cases of canine lymphoma concern middle-sized and large-breed dogs, but all dog breeds can be affected. Lymphoma may develop at any age, but it mainly occurs in middle-aged to older dogs. There is no established predisposition regarding gender, although intact female dogs appear to have a reduced risk in developing lymphoma [26]. With respect to human medicine, primary renal lymphoma occurs bilaterally in young patients and unilaterally in adults. Additionally, patients aged 0–18 years old have lower survival rate than patients between 18 and 50 years of age. It also appears that younger patients and bilateral PRL (Primary renal lymphoma) results in a shorter survival time and show more rapid progression [27]. Regarding gender, among the 16 cases of canine renal lymphoma reported, nine of them were males, five were females and for two cases data was not available. In the 14 cases, the kidneys bilaterally involved ages between 2 and 9 years old, while in a 9-year-old dog unilateral involvement was seen. The objective of this case report was to present for the first time the clinical evaluation and imaging findings, as well as cytological and pathological features of a primary bilateral renal B-cell lymphoma case in a dog.

## 2. Case Presentation

A 3-year-old intact male Rottweiler weighting 33.3 kg was admitted due to anorexia, weakness and vomiting for a three-day duration to the Companion Animal Clinic, School of Veterinary Medicine, Faculty of Health Sciences, Aristotle University of Thessaloniki. Physical examination was unremarkable except for bilateral renomegaly detected upon abdominal palpation, along with a mild bilateral cardiac murmur. Body condition score was two out of five. Complete blood count revealed neutrophilia (14 K/μL; Reference Range (RR) 3.9–8.0 K/μL) and thrombocytopenia (137 K/μL; RR 200–500 K/μL). Serum biochemistry profile showed increased urea nitrogen [BUN (Blood urea nitrogen) 96 mg/d; RR 10–38 mg/dL] and creatinine (4.9 mg/dL; RR 0.7–1.3 mg/dL) concentrations. Moreover, alkaline phosphatase (ALP) activity (186 U/L; RR 32–149 U/L), as well as phosphorus (9 mg/dL; RR 2.2–6.0 mg/dL) and total calcium (15.8 mg/dL; RR 8.6–10.9 mg/dL) concentrations were increased, while sodium concentration was decreased (140 mEq/L; RR 144–158 mEq/L). Unfortunately, it was not possible to obtain urine sample. Over the two-day hospitalization period the dog was treated with i.v. fluid administration, antibiotics, metoclopramide, and H2 receptor antagonist.

Abdominal radiographs revealed enlargement of both kidneys and reduced serosal detail (Figure 1a). Ultrasound examination of the abdomen showed abdominal effusion and enlarged kidneys exceeding 13 cm in length. There was loss of renal architecture due to the presence of renal masses (Figure 1b). Spleen appeared to have normal size and mixed echogenicity in the visceral border, indicating the presence of a mass in the splenic parenchyma. No evidence of metastasis was detected on either left or right lateral radiographs of the thorax.

Cytological examination of the abdominal fluid aspirated under ultrasound guidance revealed the presence of sporadic isolated large round cells, compatible with large lymphocytes. Moreover, fine needle aspirates obtained from both kidneys under ultrasound guidance showed multiple isolated variably sized round cells, ranging from 1.5 to 2 red blood cells in diameter, with scant intensely basophilic cytoplasm, nuclei with rough chromatin, a high nucleus to cytoplasm ratio and multiple bizarre mitotic figures. The morphological appearance is diagnostic of large cell lymphoma (Figure 2). On the second day of hospitalization, the dog’s owner complied with euthanasia due to financial restrictions in order to support his dog therapy.

Postmortem examination was carried out in the laboratory of pathology. Grossly, both kidneys appeared severely enlarged, nodular and yellowish to white in color. The distance between cranial and caudal pole measured 13 cm in both kidneys, while the distance between the renal hilus and the convex lateral border measured 9 cm and 11 cm in the left and right kidney, respectively. This significant renal enlargement occurred as a result of local tumor growths. These were numerous, large, irregularly shaped, pale and firm in consistency masses which presented both in the renal medulla and in the cortex, as well as in the renal pelvis (Figure 3a,b and Figure 4a). A few nodular structures measuring 0.5 to 1 cm in diameter randomly distributed in the spleen were also observed. The later were similar in color and texture with the renal masses (Figure 4b). The liver was enlarged and firm in consistency giving the appearance of nutmeg kernel. Lungs appeared edematous and emphysematous. Furthermore, there was dilation of the lumen and thinning of the wall of the right ventricle and also marked thickening of the left ventricle. The left atrioventricular valve was shortened with nodular thickenings (endocardosis). The surface of the aorta was roughened due to the presence of atheromatous plaques. Grayish- white granular pleural thickening horizontally arranged due to hypercalcemia was noticed. Bone marrow examination was not performed. Peripheral, thoracic and abdominal lymph nodes, including splenic, were thoroughly examined but no gross alterations were observed. Tissue samples from the kidneys and the spleen were collected and fixed in 10% neutral buffered formalin for histopathological and immunohistochemical evaluation. Histopathologically, the tumor masses in both kidneys and spleen consisted mainly of dense aggregates of round cells with high nuclear to cytoplasmic ratio and numerous mitotic figures, about 4 per high power field (Figure 5a). The above morphologic features were consistent with large cell lymphoma. Subsequently, for immunophenotyping, immunohistochemistry was performed on serial tissue sections obtained from kidneys only, using rabbit polyclonal antibodies against CD3 (103A-76, Cell Marque, Rocklin, CA, USA) and rabbit monoclonal antibodies against PAX-5 (EPR3730-2, Abcam, Cambridge, UK). Heat-induced antigen retrieval was performed with EDTA (Ethylenediaminetetraacetic acid) buffer, pH 8 for 20 min at 95 °C. Rabbit primary antibody binding was detected with goat anti-rabbit polymer HRP (Horseradish peroxidase) (ZytoChem Plus, Berlin, Germany). Color was developed with DAB substrate-chromogen system (Biogenex, Fremont, CA, USA) and tissues were counterstained with hematoxylin. CD3 antibody was used to identify T cells, Pax-5 antibody was used as a nuclear marker of B cells. Pax-5 is expressed at early stages of B cell development, in late stages of B cell differentiation, but is absent in differentiated plasma cells. CD3 staining yielded a negative reaction and Pax-5 staining was positive, indicating the diagnosis of B cell lymphoma (Figure 5b). 

## 3. Discussion

Lymphoma is one of the most common tumors in dogs over 5 years of age [19]. Canine lymphoma can affect any dog breed, but medium-sized and larger dog breeds have an increased risk of developing lymphoma, including Rottweilers [26,28]. A genetic origin or clustering has been reported in Rottweilers among other large dog breeds [26]. Moreover, some dog breeds show a significant predisposition to a specific immunophenotype of lymphoma and, particularly, Rottweillers present a higher prevalence of B-cell lymphoma [26,28].

Although renal lymphoma is usually secondary, primary renal lymphoma is infrequently described in the literature. In the veterinary literature, 16 cases of presumed primary renal lymphoma have been reported since 1954 [1,12,13,14,15,16,17,18,19,20,21,22,23,24,25]. Common nonspecific clinical signs observed in previous reports, as well as in our case, were anorexia, weight loss, vomiting, diarrhea and lethargy. Repeated haematological abnormalities, which were also observed in our case, included mild neutrophilia and thrombocytopenia while serum biochemical alterations included azotemia, hypercalcemia and hyperphosphatemia.

As seen in the present case, mild neutrophilia was observed in five cases and was likely consistent with mild inflammation [13,17,20,22,23]. Apart from our case, thrombocytopenia has been noted in just one other report [19]. Thrombocytopenia associated with neoplasia is well established and is attributed to increased platelet consumption, decreased platelet production, immune-mediated destruction, and sequestration of platelets, or a combination of all the above-mentioned disturbances [29]. Regarding serum biochemistry profile, another common finding in our case was azotemia. It was not possible to obtain urine sample and therefore we cannot know urine-specific gravity. Even if we had collected urine sample, we would not be able to evaluate it because of fluid administration. Nevertheless, our necropsy findings concerning kidneys support that renal dysfunction was the cause of azotemia. Additionally, five previous cases have reported renal azotemia [12,13,17,22,23].

In the current case, increased total calcium concentration was observed. Even though hypercalcemia would have been documented if ionised calcium was measured, the grayish- white granular pleural thickening observed at necropsy support the diagnosis of hypercalcemia. The most common cause of hypercalcemia in dogs include lymphoma (15–30% of lymphoma dogs) followed by chronic kidney disease (CKD; <5% of CKD dogs). Hypercalcemia of malignancy occurs when parathyroid-hormone related protein (PTHrP) is produced by tumor cells while renal secondary hyperparathyroidism may be its cause in CKD cases [30]. In three previous reports elevated serum calcium concentration was attributed to the production of PTHrP by the neoplastic tissue [22] or had been considered likely secondary to renal disease [12] or resulted from pseudohyperparathyroidism [13]. Furthermore, hyperphosphatemia observed in the present case probably occurred because of decreased glomerular filtration rate and/or tumor cell lysis [31]. In two previous reports, hyperphosphatemia was most likely ascribed to decreased glomerular filtration [13,17].

A common hematological finding that was not noticed in the current case, but it was repeatedly observed, was secondary polycythemia and was detected in five cases [16,17,21,23]. Appropriate secondary polycythemia is caused by increased production of erythropoietin as the result of hypoxia. Inappropriate secondary polycythemia may result from renal diseases in which excessive erythropoietin secretion occurs. Secondary polycythemia was appropriate due to presumed metastatic pulmonary disease in one case [23]. In the remainder of the cases, polycythemia was considered inappropriate as a result of erythropoietin production by neoplastic cells and/or benign renal tissue [16,17,21]. Polycythemia could be an expected finding in the present case because it occurs, among others, in dogs with renal cell carcinoma, renal lymphoma, and chronic pyelonephritis [32]. Perhaps this can be attributed to the early stage of the disease of our dog. It would be possible that cytological examination of bone marrow could give us some information about the erythroid line cellularity.

The differential diagnosis of primary renal neoplasia in dogs includes epithelial tumors (70%), mesenchymal tumors (25%), and nephroblastoma (5%) [11,33]. Although lymphoma is the most common renal neoplasm in domestic animals, it is usually of secondary origin. Regarding metastatic renal neoplasms, the most common in dogs are hemangiosarcoma, adenocarcinoma (unspecified primary), and lymphoma. Most neoplasms metastatic to the kidneys also have metastases in the lungs, but an exception to this is lymphoma, in which pulmonary tumors are rare [33].

Dogs with metastatic renal tumors initially present clinical symptoms attributed to the primary site and only rarely show significant renal dysfunction [23]. In our case clinical and laboratory findings were associated with renal disease. Primary renal lymphoma, both in humans and animals has caused controversy among researchers because of the rarity of the cases and the absence of lymphoid tissue in renal parenchyma. It is supported that primary renal lymphoma in dogs may originate from lymphatic vessels of renal capsule or from chronic inflammation of the kidney which attracts lymphocytes in the renal parenchyma and eventually are subjected to malignant transformation [19,23]. Similar pathogenesis has been described in human medicine [27]. Inflammation such as Hashimoto thyroiditis, gastritis caused by Helicobacter pylori, as well as chronic pyelonephritis may contribute to the development of lymphoma in the respective organs [34]. According to the literature, criteria proposed in humans for the diagnosis of primary renal lymphoma are the following: (i) presence of renal mass; (ii) no evidence of extrarenal lymphomatous involvement in visceral organs or lymph nodes at first admission; and (iii) absence of a leukemic blood picture together with no evidence of myelosuppression [27].

Primary splenic lymphoma (PSL) is also very uncommon, comprising less than 2% of lymphomas [35]. Several researchers have proposed a number of criteria to diagnose PSL in humans: (i) lymphoma involving only the spleen and the splenic hilar lymph nodes combined with isolated splenomegaly [36], (ii) predominance of spleen enlargement in any lymphoma involving the spleen [37], (iii) splenomegaly, cytopenia of at least two hematologic cell lines and absence of peripheral adenopathy [38] and (iv) Non Hodgkin Lymphoma (NHL) arising primarily in the spleen or as NHL principally confined to the spleen and its local lymph nodes [39].

In the current case, splenic masses were considered to be secondary tumors. This conclusion was based on the following evidence: splenomegaly and splenic lymph node involvement were absent and blood parameters were within normal limits. Finally, in contrast to the highly enlarged kidneys almost entirely occupied by the tumor, the spleen was normal in size and neoplastic nodules were sparse and measured less than 1 cm in diameter.

Conclusively, it may be suggested that the primary site of this particular lymphoma was the kidney. This was determined due to (i) the presence of multiple renal masses which were initially detected upon ultrasonography and finally confirmed during gross examination, (ii) no lymphomatous involvement of regional lymph nodes in concomitance with the identification of the small sized masses observed in the spleen as metastatic foci of lymphoma and (iii) eventually, no signs of leukemia or myelosuppression revealed in the hematology profile. As mentioned above, previous studies have supported that canine primary renal lymphoma may originate either from lymph vessels of capsule or from chronic inflammation of kidneys [19,23]. Since, neither features of renal secondary lymphoma nor of chronic inflammation were observed, we concluded that in this case lymphoma probably arose in capsular lymphatics.

In the present case, histopathology and immunohistochemistry using CD3 and PAX5 antibodies were performed. In previous reports, six cases have been diagnosed as canine renal T cell lymphoma, whereas no B-cell type case has been reported in dogs. In five of the six T-cell lymphomas the diagnosis was made by immunohistochemistry and in one with immunocytochemistry. All, of the above cases yielded a positive reaction for CD3, five of them were negative for CD79a and two cases were positive for CD18. Tissue sections from the masses found in kidneys and spleen, were negative for CD3, while nuclei of many neoplastic cells were positively stained with PAX5, leading to the diagnosis of a diffuse renal large B cell lymphoma, with metastases to the spleen. According to our knowledge this is the first case of canine renal B-cell lymphoma well established using immunohistochemistry (IHC). In the light of our findings the utility of IHC in diagnosing renal B-cell lymphomas became recognizable.

## 4. Conclusions

Diffuse large B cell lymphoma is the most common type of lymphoma in dogs. Τhe majority of the cases are high-grade lymphomas [2]. Regarding T-cell phenotype, previous studies have confirmed that it may be a prognostic factor associated with a poor outcome in both humans and dogs. Additionally, prognostic differences among the histomorphologic subtypes of B- and T-cell lymphomas have been noted [39]. Hence, taking into consideration their variations in biologic behavior, it is important to label immunohistochemically and identify the histomorphologic subtypes of lymphomas in order to provide the essential diagnostic information and subsequently an optimal treatment. Although primary renal lymphomas have rarely been reported, it is recommended, in cases where renal disease occurs due to the presence of renal neoplasia, to include primary renal lymphoma in the differential diagnosis.

## Figures and Tables

**Figure 1 vetsci-08-00258-f001:**
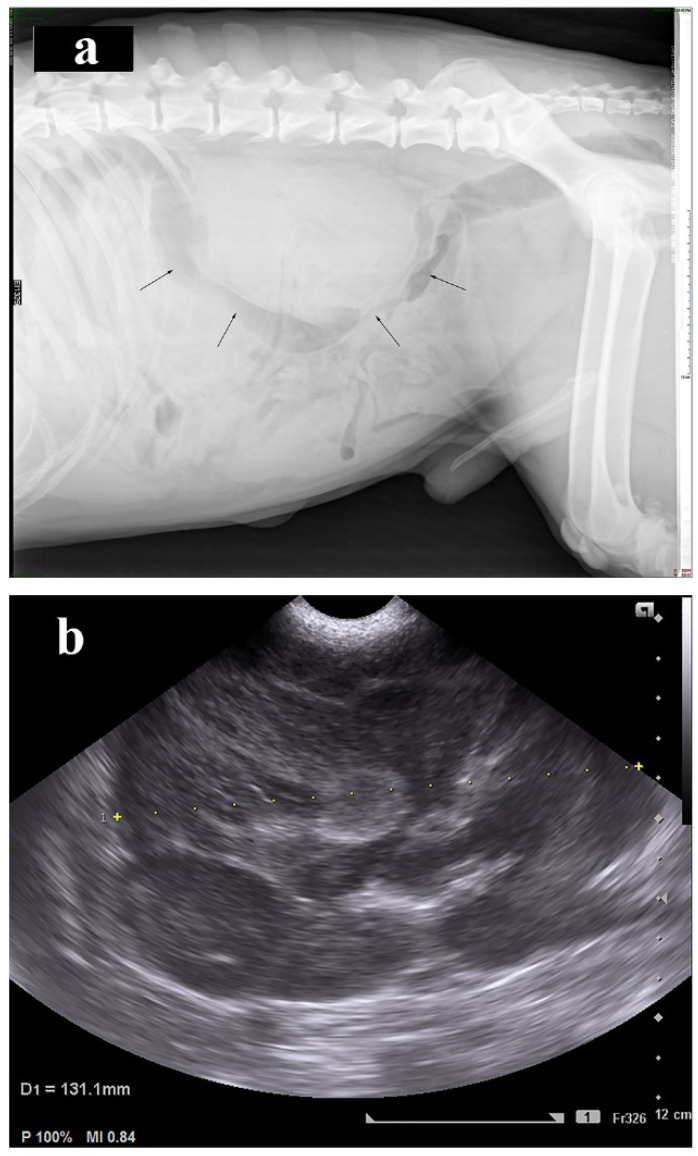
(**a**). Right lateral abdominal radiograph of a dog with renal lymphoma. Enlargement of the kidneys (arrows) with reduced serosal detail is indicated while lumbar and mesenteric lymph nodes cannot be depicted. (**b**). Ultrasound examination of the kidney of a dog with renal lymphoma. The presence of renal masses alters the normal architecture of the kidney and there is loss of corticomedullary demarcation. (+): symbolizes renal length measurement.

**Figure 2 vetsci-08-00258-f002:**
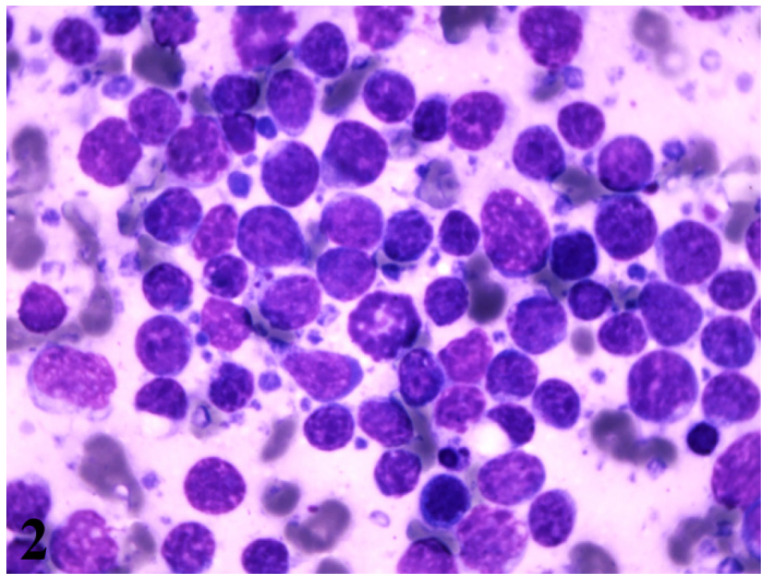
Photomicrograph of fine needle aspirate obtained from a renal mass. The majority of the cell population is composed of large lymphocytes. The background is eosinophilic and small round basophilic structures are noted (lymphoglandular bodies). The morphological appearance is diagnostic of large cell lymphoma. Renal epithelial cells are not observed. Wright Giemsa stain, oil 1000×.

**Figure 3 vetsci-08-00258-f003:**
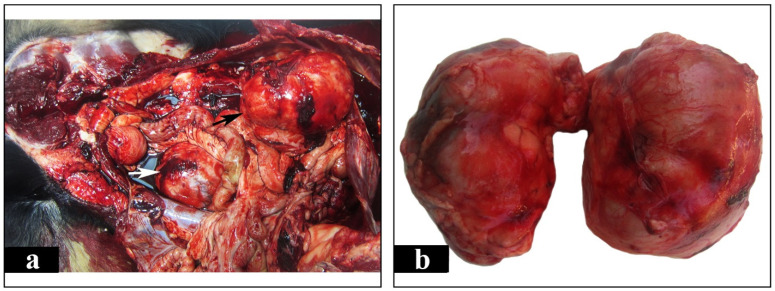
Gross appearance of the abdominal cavity and kidneys. (**a**). Both right (black arrow) and left kidney (white arrow) are intensely enlarged occupying large areas in the abdominal cavity. (**b**). Kidneys covered by their capsules, appear severely enlarged with irregular surface and mostly whitish color.

**Figure 4 vetsci-08-00258-f004:**
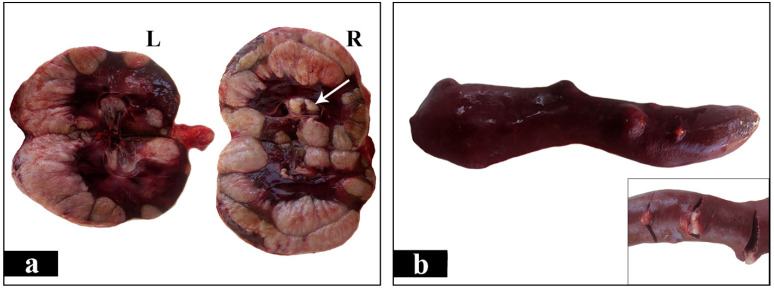
(**a**). Cut surface of kidneys show multiple whitish masses, with irregular to round shape, occupying about the ¾ and the ½ of the right (R) and the left (L) kidney, respectively. The masses are present in the medulla and the cortex, as well as in the renal pelvis (arrow). Several of them are solitary, but the majority are coalescing. (**b**). The spleen demonstrates nodules protruding from the surface. The capsule appears smooth and intact and on cut surface whitish masses similar to those observed in kidneys are obvious (inset).

**Figure 5 vetsci-08-00258-f005:**
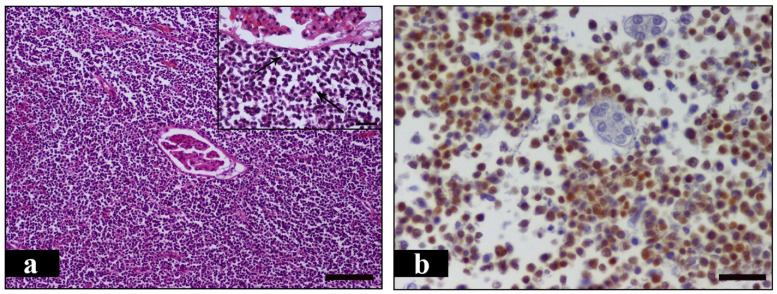
Histologic sections of a renal mass. (**a**). Numerous neoplastic mononuclear cells surround a nephron in the renal cortex. Ιnset: Tumor cells have round shape, eosinophilic cytoplasm and large nuclei while high mitotic activity is noted (arrows). H&E (Hematoxylin and eosin), bar = 100 μm, inset bar = 25 μm. (**b**). Numerous neoplastic B lymphocytes show intense positive nuclear staining for Pax-5. IHC, EnVisionTM detection, bar = 25 μm.

**Table 1 vetsci-08-00258-t001:** Data of canine renal lymphoma cases reported in the veterinary literature.

Breed	Age	Sex	Laboratory Tests	Imaging	FNA	Necropsy	Histopathology	IHC/Cell Type
Scottish Terrier	5	♂	NA	NA	NA	(+)	NA	NA
Basenji	2	♀	(+)	(+)	NA	(+)	(+)	NA
Collie	5.5	♂	NA	NA	NA	(+)	(+)	NA
Labrador	9	♂	(+)	(+)	NA	NA	(+)	NA
Unknown	NA	NA	NA	(+)	NA	NA	(+)	NA
Medium or large breed	NA	NA	(+)	(+)	NA	(+)	(+)	NA
ShibaInu	3	♀	(+)	(+)	NA	(+)	(+)	NA
Doberman	5	♀	(+)	NA	NA	NA	(+)	NA
Staffordshire	5	♀	(+)	NA	NA	(+)	(+)	(+)/T
Basset Hound	8	♂	(+)	(+)	NA	NA	(+)	(+)/T
Springer Spaniel	8	♂	(+)	(+)	(+)	(+)	(+)	NA
Flat coated Retriever	6	♂	(+)	(+)	(+)	NA	NA	NA
Mixed breed	2	♂	(+)	(+)	NA	(+)	(+)	(+)/T
Cocker Spaniel	3	♂	(+)	(+)	NA	(+)	(+)	(+)/T
Border Collie	9	♀	(+)	(+)	NA	(+)	(+)	(+)/T
Bernese mountain	8	♂	(+)	(+)	(+)	NA	NA	(+)/T

(+) provided data; NA: not available; ♀ female; ♂ male; T: T-cell lymphoma. FNA: Fine needle aspiration, IHC: Immunohistochemistry.

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
