# Peer review of "Bilateral Renal Large B Cell Lymphoma in a Dog: A Case Report and Review of the Literature"

_vetsci, 2021, doi:10.3390/vetsci8110258_

Round 1

Reviewer 1 Report

Comments to the Author

        In this manuscript, the authors report the clinic evaluation and imaging findings, cytological and histological findings and immunohistochemical features of a canine bilateral renal large B cell lymphoma and compare with previous primary renal lymphoma. It is an interesting study from the surgical pathology and comparative pathology aspect. The main weaknesses of this study are shortage of discussion. Although the title of the report that is a case report and review of the literature, there is a lack of information on how the gross and histological findings differ from previous reports. It would be good to included in the discussion that compare this case with the previous reports.

Minor comments: 

  1. Line 42: autopy >> necropsy
  2. Lines 50-51: There is insufficient comparison in human medicine. Please describe the detail the human primary renal lymphoma.
  3. Line 79: ‘It was not possible to collect urine sample.’ is duplicate with Line 72-73.
  4. Lines 106, 111: “9cm”, “11cm” and “1cm” Please add a space between number and cm.
  5. Lines 123-125: Is the histological findings correct to that the same findings were observed in the spleen? Please describe the findings in the kidney and spleen separately.
  6. Lines 129-130: Please mention treated temperature and time.
  7. Line 157: replace ‘clustering’ with ’cluster’.
  8. Line 260: Did you performed immunohistochemistry using CD79a?
  9. Figure 2: There is not insert a scale in the figure.

Author Response

Author's Reply to the Review Report (Reviewer 1)

Line 42: autopy >> necropsy

Replaced in the manuscript

Lines 50-51: There is insufficient comparison in human medicine. Please describe the detail the human primary renal lymphoma.

We hope that the information we added in the revised manuscript will meet the reviewers’ requirements

Line 79: ‘It was not possible to collect urine sample.’ is duplicate with Line 72-73.

Corrected.

Lines 106, 111: “9cm”, “11cm” and “1cm” Please add a space between number and cm.

Corrected.

Lines 123-125: Is the histological findings correct to that the same findings were observed in the spleen? Please describe the findings in the kidney and spleen separately.

As the reviewers can see in the revised manuscript histological findings in kidneys and spleen were the same.

Lines 129-130: Please mention treated temperature and time.

The answer to reviewers’ request has been included in the revised manuscript.

Line 157: replace ‘clustering’ with ’cluster’.

We used the term “clustering” according to the relevant reference [26].

Line 260: Did you performed immunohistochemistry using CD79a?

We did not use CD79a antibody.

Figure 2: There is not insert a scale in the figure.

Corrected.

Reviewer 2 Report

Dear Authors,

I found your case report interesting and well written, and it is important to have added a review of the literature. It is obvious that there is no particular scientific discovery at the basis, but the purpose of describing a case in an exhaustive way is achieved.

I have only a few small observations:

  • line 92 would indicate the size of the neoplastic cells by comparing them to red blood cells. Was there any mitosis?
  • the photos: for example, figure 5 is a bit dark for me, I would do a strong enlargement of the immunohistochemistry photo 5b.
  • line 252: how do you claim that capsular lymphatics are the origin of the neoplasm? are there bibliographic data? I would be more generic.

Author Response

Author's Reply to the Review Report (Reviewer 2)

line 92 would indicate the size of the neoplastic cells by comparing them to red blood cells. Was there any mitosis?

The answer to reviewers’ request has been included in the revised manuscript.

the photos: for example, figure 5 is a bit dark for me, I would do a strong enlargement of the immunohistochemistry photo 5b.

We replaced figure 5b with higher magnification.

line 252: how do you claim that capsular lymphatics are the origin of the neoplasm? are there bibliographic data? I would be more generic.

We hope that the information we added in the revised manuscript will meet the reviewers’ requirements.